# Women's traditional birth attendant utilization at birth and its associated factors in Angolella Tara, Ethiopia

**Birhan Tsegaw Taye**[1]*, **Mulualem Silesh Zerihun**[1], **Tebabere Moltot Kitaw**[1], **Tesfanesh Lemma Demisse**[1], **Solomon Adanew Worku**[1], **Girma Wogie Fitie**[1], **Yeshinat Lakew Ambaw**[1], **Nakachew Sewnet Amare**[1], **Geremew Kindie Behulu**[1], **Addisu Andualem Ferede**[2], **Azmeraw Ambachew Kebede**[3]

1 School of Nursing and Midwifery, Asrat Woldeyes Health Science Campus, Debre Berhan University, Debre Berhan, Ethiopia, 2 Department of Midwifery, College of Medicine and Health Science, Debre Markos University, Debre Markos, Ethiopia, 3 Department of Clinical Midwifery, School of Midwifery, College of Medicine and Health Sciences, University of Gondar, Gondar, Ethiopia

* tsegawbirhan2@gmail.com

## Abstract

### Background

In developing countries, home delivery without a skilled birth attendant is a common practice. It has been evidenced that unattended birth is linked with serious life-threatening complications for both the women and the newborn. Institutional delivery with a skilled birth assistance could reduce 20–30% of neonatal mortality. This study aimed to assess traditional birth attendants' (TBAs) utilization and associated factors for women who gave birth in the last two years in Angolella Tara District, Ethiopia.

### Methods

A community-based cross-sectional study was employed among 416 women who gave birth in the last two years at rural Angolella Tara District. Study participants were recruited by using a simple random sampling technique. Data were collected using a structured, pre-tested, and interviewer-administered questionnaire. Epi Data 4.6 and SPSS version 25 were used for data entry and analysis, respectively. A multivariable logistic regression model was fitted to identify factors associated with women's utilization of traditional birth attendants. The level of significance in the last model was determined at a p-value of <0.05.

### Result

Overall, 131 (31.5%) participants were used traditional birth attendants in their recent birth. Unmarried marital status (AOR 2.63; 95% CI: 1.16, 5.97), age at first marriage (AOR 2.31; 95%CI: 1.30, 4.09), time to reach health facility (AOR 3.46; 95% CI: 1.94, 6.17), know danger sign of pregnancy and childbirth (AOR = 5.59, 95% CI; 2.89, 10.81), positive attitude towards traditional birth attendants (AOR = 2.56 95% CI; 1.21,5.52), had antenatal care

**Data Availability Statement:** All relevant data are within the paper and its Supporting Information files.

**Funding:** The author(s) received no specific funding for this work.

**Competing interests:** The authors have declared that no competing interests exist.

**Abbreviations:** ANC, Antenatal Care; MMR, Maternal Mortality Ratio; MOH, Ministry of Health; MWH, Maternity Waiting Homes; PPH, Post-Partum Hemorrhage; SBA, Skilled birth attendant; SDG, Sustainable Development Goal; TBA, Traditional Birth Attendance; WHO, World Health Organization.

follow-up (AOR: 0.11 95% CI 0.058, 0.21), and listening radio (AOR = 0.43; 95% CI: 0.18, 0.99) were significantly associated factors with the use of traditional birth attendants.

## Conclusion

Nearly one-third of women used traditional birth attendant services for their recent birth. TBAs availability and accessibility in the community, and respect for culture and tradition, problems regarding infrastructure, delay or unavailability of ambulance upon call, and some participants knowing only TBAs for birth assistance were reasons for preference of TBAs. Therefore, effort should be made by care providers and policymakers to ensure that modern health care services are accessible for women in a friendly and culturally sensitive manner. In addition, advocacy through mass media about the importance of maternal health service utilization, particularly antenatal care would be important.

## Introduction

The World Health Organization (WHO) defines traditional birth attendant (TBA) as a person who assists the mother during labor and delivery, and who acquired her skills by delivering babies herself or trainings of other TBAs [1, 2]. TBAs are involved in 60–80% of all deliveries in rural regions in low resource settings [3].

Global statistics show that more than 303 000 women die annually 2.6 million babies were stillborn, half occurring during the third trimester [4] with the vast majority (94%) of these deaths occurring in low resource settings [5]. From the global figure, approximately 14,000 maternal deaths occur each year in Ethiopia with a higher lifetime risk of maternal mortality [6]. The current evidence shows that four women die from every 1000 live births in Ethiopia, which is the highest maternal death in the world [7].

This higher maternal mortality is as a result of unavailable, inaccessible, costly, or poor quality maternal healthcare services [8–10].

Skilled care during the maternal continuum of care can save the lives of both the women and newborns [6, 11]. Improving access to skilled care throughout the maternal continuum of care is a top priority to improve maternal health and meet the sustainable development goals (SDG3) [12]. Since the launching of the Health Extension Program in 2003, there has been a change in Ethiopian health policy that TBAs are no longer allowed to attend births [13]. However, most women in developing countries give birth outside of health facilities with the help of TBAs [10, 14]. In 2017, the MMR in low-income nations was 462/100,000 live births, which is much higher compared with the high-income countries (i.e., 11/100,000 live births) [14]. Although pregnancy related complications are associated with inability care from a skilled professional, low socioeconomic status, number of living children, married predominantly farmers, married to spouses who were farmers without formal education, friendly care from TBAs, easily accessible, delivery environment, and religious beliefs of clients were also significant factors [10, 15–20].

Ethiopian mini demographic health survey 2019 (EMDHS 2019) report showed that only 50% of births were attended by skilled birth attendants (SBAs) [21]. The Ethiopian government was planned to increase the proportion of SBAs to 90% by 2020, but not achieved yet. Hence, home birth rates are much higher in rural parts of the country than in urban areas (urban: 29.6% vs. rural: 60%) [22] Also, a significant proportion of women was not fully

engaged in the continuum of maternity care with significant dropouts [23]. For instance, 74% of pregnant women undergo at least one antenatal care (ANC) visit, 48% of women gave birth in a health institution, and 34% of women received postnatal care in the first 2 days after birth [22]. Though there is a need to improve the utilization of health facility deliveries with SBAs, many women gave birth at home with the aid of TBAs in rural and deprived communities of Ethiopia. Thus, evidence regarding women's attitude towards TBAs and reasons for preference for TBAs were lacking so far. Therefore, this study aimed to assess the reasons for traditional birth attendants' preference, utilization, and associated factors among women who gave birth in the last two years in Angolella Tara District, Ethiopia.

## Methods and materials

### Study design, period, and setting

A community-based cross-sectional study was conducted from June 1st to 15th, 2021. This study was conducted in the rural Angolella-Tara district, Amhara regional state, Northeast Ethiopia. The district is located at the eastern edge of the Ethiopian highlands in the North Shewa zone, 112 km Northeast of Addis Ababa (the capital city of Ethiopia). It is called in part after one of the capitals of the former principality of Shewa, Angolella Tara district. Currently, the district has 21 kebeles with an area of 782.49 km$^2$ and a total estimated population of 82,349, of whom 40,500 were women (based on the 2007 national census). Also, there are four health centers and 21 health posts serving the community.

### Source population and study population

All women who gave birth in the last two years in the Angolella-Tara district were the source populations. All women who gave birth in the last two years in selected kebeles at Angolella -Tara district during the data collection period were the study population.

### Eligibility criteria

**Inclusion criteria.** All women who gave birth in the last two years and permanently residing (6+ months) in selected kebeles at Angolella -Tara district during the data collection period were included.

**Exclusion criteria.** Women who were seriously ill and not available at their homes in repeated visits of the data collection period were excluded.

### Sample size determination

The sample size for this study was determined by using a single proportion formula by considering the following assumptions: a 50% proportion of women's utilization of TBAs, 95% level of confidence, and a 5% margin of error.

$$N = \frac{(Z\alpha/2)^2 p(1-p)}{d^2} \text{Therefore}, n = \frac{(1.96)^{2*}0.5(1-0.5)}{(0.05)2} = 384.$$

By considering a 10% non-response rate, the minimum adequate sample size was 422.

### Sampling technique and procedure

From the total of 21 kebeles, nine kebeles were selected randomly using a lottery method. The list of the study participants was gained from health extension workers (HEWs) and local administrators. The sampling frame was designed by numbering the list of women. Then, the

total sample size was distributed to each selected "kebeles" proportional to the size. Lastly, the eligible women were chosen by a simple random sampling technique.

## Variables of the study

**Dependent variable.** Utilization of TBAs as birth attendants (Utilized/not Utilized).

**Independent variables.** **Socio-demographic characteristics** (Maternal age, Maternal occupation, Husband occupation, Mother's educational status, Husband's educational status, Media exposure, Income, Marital status, Family size, listening to radio, reading newspaper).

**Reproductive health-related characteristics** (Parity, Family planning, planned pregnancy, Assistant of the recent delivery, Knowledge of obstetric danger signs, having ANC visit, Family size, Alive children, Healthcare service decision making, knowing maternity waiting home); and Attitude towards TBAs.

## Measurement and operational definitions

### Traditional birth attendant

A TBA is defined by the WHO as a person, generally a woman, who assists pregnant women during childbirth and who learned the skills by attending homebirths, potentially with the help of other TBAs [1, 21].

### Skilled attendant

A professionally trained healthcare provider having the essential skills to manage normal labor and delivery, recognize complications early and perform any essential interventions including early referral [24].

### Utilization of traditional birth attendants

When women are delivered at home or birth that takes place in a residence without a SBA; the possible answers were Yes or No. A score of "1" was given for Yes and a score of "0" was given for No which was dichotomized as utilized and not utilized [2, 13].

### Women's attitude

Women's attitude towards TBAs was measured using 15 questions: Each question has five points Likert scale (1 = strongly disagree, 2 = disagree, 3 = neutral, 4 = agree, 5 = strongly agree). The total score was 15–75 and women who scored 50% and above value were considered as having a positive attitude and those who scored below 50% were considered as having a negative attitude.

### Know obstetric danger signs

Women who were able to list at least two of the key danger signs (vaginal bleeding, swelling of face, fingers, severe persistent abdominal pain, blurring of vision, severe recurrent frontal headache, high-grade fever, and swelling of the face) categorized know the danger signs and no otherwise [25, 26].

### Planned pregnancy

A woman who plans to become pregnant by making lifestyle choices for optimal health in advance of the planned conception.

## Maternity waiting home

If mothers are living far from the delivery center, they shall be admitted to a maternity waiting home (residential facility) which is located near or within hospitals in their final weeks of pregnancy to bridge the geographical gap in obstetric care [27, 28].

## Data collection tools and quality control

Data were collected using a pre-tested, structured, and interviewer-administered questionnaire through face-to-face interviews. The questionnaire was prepared by reviewing related literature. The questionnaire was first prepared in English and translated into Amharic (local language), and then back to English to maintain the consistency of the questionnaire. The face and content validity of the questionnaire was validated by public health experts. Reliability was checked and Cronbach's alpha coefficients was 0.87. The data collection and supervision were conducted by six Bachelor of Science and one MSc midwives, respectively. A pretest was done at Baso district on 5% of the sample size to check language clarity, and appropriateness of the questionnaire. Based on the pretest, necessary amendments were done before the actual data collection.

## Data processing and analysis

All information was recorded corresponding to the code given to each respondent. Data were checked, coded, and entered into EpiData version 4.6, and were exported to SPSS version 25 for analysis. Descriptive statistics were used to present the participants' background information. Both bivariable and multivariable logistic regressions analysis were employed. Variables with a p- value of ≤0.25 on the bivariable logistic regression were selected as a candidate variable for the multivariable logistic regression. The level of significance was determined at a p-value of <0.05 and the strength of association was interpreted using the adjusted odds ratio (AOR) with its 95% confidence interval (CI).

## Ethical consideration

The study was conducted under the Ethiopian Health Research Ethics Guideline and the declaration of Helsinki. Ethical clearance was obtained from Debre Berhan University ethical committee (protocol number P09/21). A formal letter of administrative approval was gained from the Angolella Tara district and local administrator. Written informed consent was obtained from respondents before data collection by clarifying the objective of the study. The name of participants was not written and confidentiality was maintained throughout the study. The data collector explained that the respondent can withdraw from the study any time and that participation was voluntary.

## Result

### Sociodemographic characteristics

A total of 416 women participated in this study, giving a response rate of 98.58%. A bit more than half (51.2%) of the study participants was in the age group of 26–30 years and the mean age of the study participants was 33.56 years (SD ±8.01). The majority (86.8%) of the study participants were married and almost all (99.3%) of them were Orthodox Christian by religion [Table 1].

**Table 1. Socio-demographic characteristics of women in Angolella Tara District Northeast Ethiopia, 2021 (n = 416).**

| Characteristics | Frequency | Percentage (%) |
|---|---|---|
| **Age** | | |
| ≤25 | 29 | 7.0 |
| 26–30 | 213 | 51.2 |
| >30 | 174 | 41.8 |
| **Current marital status** | | |
| Single | 55 | 13.2 |
| Married | 361 | 86.8 |
| **Participant level of Education** | | |
| No formal education | 294 | 70.7 |
| Primary (1–8) | 95 | 22.8 |
| Secondary (9–12) | 12 | 2.9 |
| College and above | 15 | 3.6 |
| **Husband/friend level of education (n = 361)** | | |
| No formal education | 239 | 66.21 |
| Primary (1–8) | 66 | 18.28 |
| Secondary (9–12) | 43 | 11.9 |
| College and above | 13 | 3.61 |
| **Read newspaper** | | |
| Yes | 47 | 11.3 |
| No | 369 | 88.7 |
| **How often read a newspaper** | | |
| Always | 5 | 15.1 |
| Once/week | 27 | 64.2 |
| Twice/week | 8 | 9.4 |
| Three and above/week | 7 | 11.3 |
| **Listen to radio?** | | |
| Yes | 77 | 18.5 |
| No | 339 | 81.5 |
| **How often listen radio** | | |
| Always | 19 | 54.8 |
| Once | 14 | 9.1 |
| Twice/week | 16 | 8.1 |
| Three and above/week | 28 | 27.9 |
| **Participant occupation** | | |
| Housewife | 305 | 73.3 |
| Self-employed | 19 | 4.6 |
| Merchant | 62 | 14.9 |
| Private/government employee | 30 | 7.2 |
| **Husband/friend occupation (n = 361)** | | |
| Farmer | 280 | 77.56 |
| Self-employed | 70 | 19.39 |
| Merchant | 11 | 3.05 |
| Private/government employed | 16 | 4.43 |
| **Religion** | | |
| Orthodox | 413 | 99.3 |
| Muslim | 3 | 0.7 |

*(Continued)*

**Table 1.** (Continued)

| Characteristics | Frequency | Percentage (%) |
|---|---|---|
| **Average monthly income in Ethiopian Birr** | | |
| < 3000 | 132 | 54.6 |
| 3001–5000 | 227 | 31.7 |
| >5000 | 57 | 13.7 |

Note
*physicians

## Reproductive health service-related characteristics of women

More than half of women (53.4%) have ever used family planning methods. About 76.9% of the pregnancies were unplanned. Two hundred sixty (62.5%) of women were assisted by a SBA and only 25(6%) were assisted by their family members for their most recent birth. Moreover, 299 (71.9%) of the respondents had ANC follow-ups in the preceding pregnancy (71.4%) [Table 2].

## Utilization and attitude of women towards TBAs

Overall, 131(31.5%; 95% CI: 27%, 36%) women used TBAs services for their most recent childbirth. More than half (41.8%) of study participants have a positive attitude towards TBAs.

## Women's reason for preference for traditional birth attendants

There are many reasons for women's preference for traditional birth attendants. Among these, TBAs are available in rural areas (36.54%), transportation access problems (19.47%), respect of culture and tradition (16.35%), they know only TBAs (14.90%), and unavailability and delay of ambulance upon call (12.74%) [Fig 1].

## Factors associated with women's utilization of traditional birth attendants

Multivariable logistic regression analysis revealed that being unmarried, age at first marriage, ANC follow-up, time to reach a health facility, listening to radio, didn't know obstetric danger signs, and positive attitude towards TBAs were factors associated with TBAs service utilization.

Accordingly, this study found that those women with unmarried current marital relations were 2.63 (AOR = 2.63; 95% CI: 1.16, 5.97) times more likely to use TBAs as compared to married women. Those mothers whose age at first marriage was less than 18-year-old were 2.31 (AOR = 2.31; 1.30, 4.09) times more likely to use TBA services. This study also revealed that women who traveled for greater than an hour were 3.46 times more likely to have TBAs compared with those women who traveled for less than an hour (AOR = 3.46; 95% CI: 1.94, 6.17). Mothers who didn't know the danger signs of pregnancy and childbirth were 5.59 times more likely to utilize TBAs than their counterparts (AOR = 5.59; 95% CI: 2.89, 10.81).

Moreover, mothers who have a positive attitude towards TBAs were 2.23 times more likely to use TBAs than those who have a negative attitude AOR = 2.56; 95% CI: 1.21, 5.52). However, women who have ANC follow-up were 89% less likely to use TBA services during birth. Likewise, participants who listen to the radio at least once per week were 57% less likely utilizing TBAs for their childbirths as compared to their counterparts (AOR = 0.43; 95% CI: 0.18, 0.99) [Table 3].

**Table 2. Reproductive health service-related characteristics of women in Angolella Tara District of Northeast Ethiopia, 2021 (n = 416).**

| Characteristics | Frequency | Percentage (%) |
|---|---|---|
| **Ever use family planning methods** | | |
| Yes | 222 | 53.4 |
| No | 194 | 46.6 |
| **Pregnancy planned** | | |
| Yes | 96 | 23.1 |
| No | 320 | 76.9 |
| **Have ANC in preceding pregnancy** | | |
| Yes | 299 | 71.9 |
| No | 117 | 28.1 |
| **Know obstetric danger signs** | | |
| Yes | 196 | 47.1 |
| No | 220 | 52.9 |
| **Age at first marriage** | | |
| <18 | 166 | 39.9 |
| ≥18 | 250 | 60.1 |
| **Parity** | | |
| 1 | 60 | 14.4 |
| 2–4 | 200 | 48.1 |
| ≥5 | 156 | 37.5 |
| **Number of live children** | | |
| ≤2 | 115 | 27.6 |
| 3–4 | 148 | 35.6 |
| ≥5 | 153 | 36.8 |
| **History of medical illness** | | |
| Yes | 41 | 9.9 |
| No | 375 | 90.1 |
| **Know the presence of maternal waiting homes** | | |
| Yes | 162 | 38.9 |
| No | 254 | 61.1 |
| **Recent delivery assisted by** | | |
| Skilled healthcare provider | 260 | 62.5 |
| Traditional birth attendant | 131 | 31.5 |
| Family member | 25 | 6.0 |
| **Owns healthcare provider decided by** | | |
| Me only | 20 | 4.8 |
| Husband/partner | 16 | 3.8 |
| Family member | 83 | 20.0 |
| Jointly | 297 | 71.4 |

ANC = Antenatal care

## Discussion

Utilization of maternal health services by skilled health professional throughout the continuum of care is the pillar in reducing MMR. However, Ethiopia has a higher proportion of homebirth (73.5%) with significantly lower health facility delivery [29]. In recent years, the need for TBA in maternal health care is in debate [30]. Opponents of TBA care are of the view

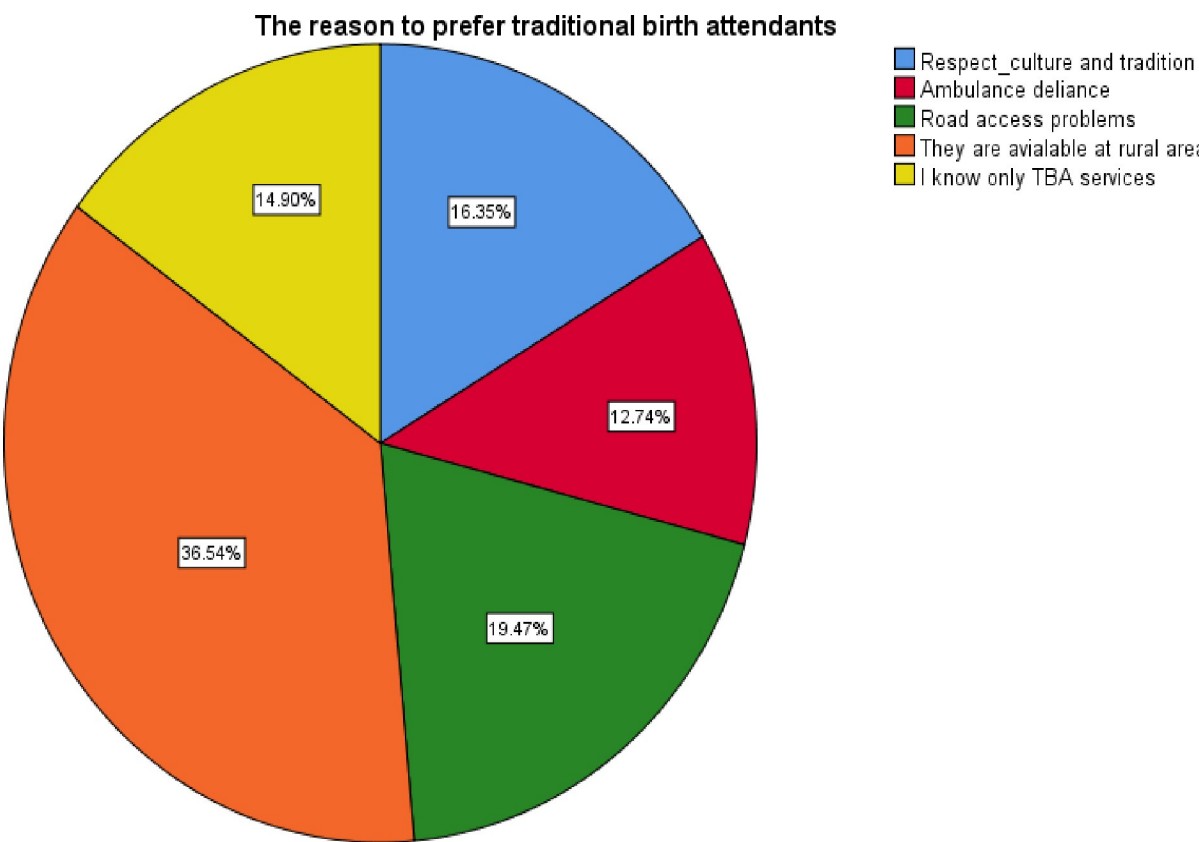

**Fig 1. Distribution of reasons for women preference on the use of traditional birth attendants (TBAs) other than skilled birth attendants (SBAs) in Angolella Tara District, Northeast Ethiopia 2021.**

that TBAs have done little to improve maternal health. After the advent of the WHO Safe Motherhood initiative, the enthusiasm turned away from TBAs [31] as they are unskilled and unable to prevent or treat the complications during pregnancy or childbirth that leads to maternal deaths [32]. This study was assessed reasons for TBAs preference, utilization, and associated factors for women who gave birth in the last two years. Thus, 31.5% of women were assisted by TBAs in their recent birth, which is lower than the study done in Nigeria (65%) [33]. The possible explanation could be the Nigerian government encourages the integration of TBAs into the health sector to improve maternal health [34].

The current study found that the time taken to the nearby health facility was significantly associated with the use of TBAs for childbirth. It has been shown that women who travel for greater than an hour to reach health facilities were 3.46 times high likely to attend their birth by TBAs as compared to those women traveling for less than one hour. This finding is consistent with studies conducted in Ethiopia [35], Eretria [36], Kenya [37], and Zambia [38]. This could be due to the fact that distance to a health facility is one of the well-known barriers to healthcare access especially in developing countries [39, 40].

The present study also showed that having ANC follow-up had a significant association with the use of TBA services. Thus, women who had ANC follow-up for their recent pregnancy were 89% less likely assisted by TBAs. This finding is supported by the study conducted in Kenya, in which women who have no ANC follow-up pose an increased utilization of TBAs [41]. This could be justified by ANC is better to decrease those problems by filling the gap in

**Table 3. Bi-variable and multivariable logistic regression analysis factors associated with the use of traditional birth attendants by women in Angolella Tara District of Northeast Ethiopia, 2021 (n = 416).**

| Variables | TBA utilization | | COR (95%CI) | AOR (95%CI) | P-value |
|---|---|---|---|---|---|
| | Yes | No | | | |
| **Age in year** | | | | | |
| 18–24 | 4 | 25 | 0.26 (.085, 0.77) | 0.68 (0.13, 3.64) | 0.984 |
| 25–34 | 60 | 153 | 0.63 (0.41, 0.96) | 0.71 (0.62, 2.28) | 0.090 |
| ≥ 35 | 67 | 107 | 1 | | |
| **Current marital status** | | | | | |
| Single | 20 | 35 | 1.29 (0.71, 2.33) | 2.63 (1.16, 5.97) | 0.015 |
| Married | 111 | 250 | 1 | | |
| **Maternal occupation** | | | | | |
| Housewife | 113 | 192 | 8.24 (1.93, 35.24) | 5.77(0.76, 33.66) | 0.110 |
| Self-employed | 2 | 17 | 1.65 (.21, 12.8) | 1.24(0.10, 15.05) | 0.736 |
| Merchant | 14 | 48 | 4.08 (0.86, 19.3) | 10.78(1.0, 60.89) | 0.051 |
| Private/gov't employed | 2 | 28 | 1 | 1 | |
| **Age at first marriage** | | | | | |
| <18 | 79 | 87 | 3.46 (2.25, 5.32) | 2.31 (1.30, 4.09) | 0.015 |
| ≥18 | 52 | 198 | 1 | | |
| **Pregnancy planned** | | | | | |
| Yes | 83 | 237 | 0.082 (0.24 0, 0.61) | 0.93 (.39, 2.23) | 0.925 |
| No | 48 | 48 | 1 | | |
| **Have ANC follow-up** | | | | | |
| Yes | 47 | 252 | 0.07 (0.04, 0.12) | 0.11(.058, 0.21) | 0.000 |
| No | 84 | 33 | 1 | | |
| **Number of live children** | | | | | |
| ≤2 | 20 | 95 | 0.29 (0.16, 0.51) | 0.88 (0.24, 3.26) | 0.69 |
| 3–4 | 46 | 102 | 0.61(0.38, 0.98) | 1.55 (0.53, 4.56) | 0.67 |
| ≥5 | 65 | 88 | 1 | | |
| **Parity** | | | | | |
| 1 | 9 | 51 | 0.24 (0.11, .52) | 0.43 (0.15, 1.29) | 0.64 |
| 2–4 | 56 | 144 | 0.53 (0.34, 0.83) | 1.02 (0.53, 1.97) | 0.40 |
| ≥5 | 66 | 90 | 1 | | |
| **Family size** | | | | | |
| <3 | 6 | 15 | 1.0 (0.29, 2.18) | 1.63 (0.44, 5.99) | 0.57 |
| 3–5 | 45 | 141 | 1.55 (0.58, 4.16) | 1.59 (0.74, 3.43) | 0.56 |
| >5 | 80 | 129 | 1 | | |
| **Listening radio at least once per week** | | | | | |
| Yes | 11 | 66 | 0.30 (0.16, 0.60) | 0.43 (0.18, 0.99) | 0.048 |
| No | 120 | 219 | 1 | | |
| **Read newspaper at least once per week** | | | | | |
| Yes | 6 | 41 | 1 | | |
| No | 125 | 244 | 3.50 (1.45, 8.47) | 1.48 (0.14, 1.65) | 0.275 |
| **Time to reach a health facility** | | | | | |
| < 60 minutes | 38 | 204 | 1 | | |
| ≥60 minute | 93 | 81 | 6.16 (3.9, 9.73) | 3.46 (1.94, 6.17) | 0.000 |
| **Ever use family planning methods** | | | | | |
| Yes | 31 | 191 | 1 | | |
| No | 100 | 94 | 6.56 (4.09, 10.51) | 1.72 (0.82, 3.6) | 0.284 |

*(Continued)*

**Table 3.** (Continued)

| Variables | TBA utilization | | COR (95%CI) | AOR (95%CI) | P-value |
|---|---|---|---|---|---|
| | Yes | No | | | |
| **Know obstetric danger signs** | | | | | |
| Yes | 18 | 178 | 1 | | |
| No | 113 | 107 | 10.44 (6.0, 18.14) | 5.59(2.89,10.81) | 0.000 |
| **Ever faced obstetric danger signs** | | | | | |
| Yes | 27 | 168 | 1 | | |
| No | 104 | 117 | 5.53 (3.41, 8.98) | 1.93(0.41, 2.13) | 0.26 |
| **Assistant for the recent delivery** | | | | | |
| SBA | 73 | 187 | 1 | | |
| TBA | 36 | 95 | 0.97 (0.61, 1.55) | 0.93 (0.49, 1.78) | 0.94 |
| Family member | 22 | 3 | 18.79(5.46,64.67) | 3.52 (0.75, 16.56) | 0.11 |
| **Know the presence of MWHs** | | | | | |
| Yes | 32 | 130 | 0.39 (0.24, 0.61) | 0.35 (0.18, 1.09) | 0.07 |
| No | 99 | 155 | 1 | | |
| **Attitude towards TBAs** | | | | | |
| Negative | 56 | 186 | 1 | | |
| Positive | 75 | 99 | 2.52 (1.65, 3.84) | 2.23 (1.24, 4.02) * | 0.005 |

Note: MWHs = maternity waiting homes; TBAs = traditional birth attendants; SDA = Skilled birth attendant

the continuum of care through early diagnosis, treatment, and preventing pregnancy-related problems.

Women's first marriage was a significant risk factor for TBA utilization. In this regard, women who were married before or at the age of 18-year-old were 2.31 times more likely to use TBA in their current childbirth as compared to their older counterparts. This finding is in agreement with studies conducted in Kenya [37], and Zambia [38]. This could be because experience in pregnancy and pregnancy-related complications increases as women's age advances which could help them to visit health facilities, thereby obtaining a comprehensive and favorable awareness of maternal health services in the subsequent pregnancy [41, 42]. Besides, older women could have a higher decision-making power regarding their health and health care utilization as compared to their younger counterparts [43].

The current study also revealed that the odds of having TBA utilization among women who had a positive attitude were 2.23 times higher as compared to those women who had a negative attitude towards TBAs. This finding is consistent with previous studies conducted in Ethiopia [44, 45]. This may be due to lack of knowledge and awareness, traditional views, user friendly, and confidence in TBAs; closer collaboration with TBAs may provide a suitable platform in the community [15]. Hence, appropriate interventions that can improve women's attitude towards SBAs.

Moreover, this finding indicated that those women who didn't know obstetric danger signs were 5.59 times more likely to use TBAs during their childbirth. This suggests that women who are pregnant for the first-time use TBA services more as they may not have awareness for obstetric danger signs, complications, and the benefits of SBAs. Consequently, they could get information regarding the danger signs of pregnancy and the importance of utilizing SBAs for subsequent pregnancies.

The odds of TBA service utilization among women who listen to radio at least once per week were less likely to approve TBAs services utilization. The possible explanation could be

due to the reality that information, education, and communication regarding maternal, neonatal, and child health improvement could be disseminated through different mass media [44, 46].

Importantly, this study examined the reasons for the preference for TBAs over SBAs. Thus, the major reasons why women prefer TBAs other than SBAs are: "TBAs are available and accessible in a rural area, roads access problems, TBAs respect culture and tradition, and there is a delay of an ambulance". Even in the presence of health professionals and HEWs, TBAs are the first choice of women in the rural community as services are provided within the home and involved in TBAs tailored to the community's norms [15]. This could be the reason why most women in rural communities build trust in TBAs. Lack of trust in the HEWs' capacity and limited coordination contribute to TBAs being the choice of birth attendants [47–49].

This finding calls for policymakers like the Ethiopian Ministry of Health (MoH) and other stakeholders should strengthen health extension programs based on the assumption that access to and quality of primary healthcare in rural communities can be improved through the transfer of health knowledge, attitude, and skills to households.

## Strengths and limitations of the study

We are pleased to acknowledge some of the limitations of the current study. First, as the area is not well studied, we didn't find adequate studies to compare and contrast our findings with others, which made our discussion shallow. Second, due to the cross-sectional nature of the study design, the cause-effect relationship between the outcome and explanatory variables might be impossible. Third, it is impossible to couch the validity of the responses that social desirability bias may be introduced. Despite these limitations, this study provides valuable information about women's TBA utilization, factors associated with it. It also supplied us a better understanding of the reasons for preference for TBAs among rural population.

## Conclusion

The extent of TBA service utilization among women who gave birth in the last two years was high in the study setting. marital status, age at first marriage, time to reach a health facility, knowing danger signs of pregnancy and childbirth, attitude towards traditional birth attendants, ANC, and listening radio were significant factors for TBAs service utilization. Additionally, TBAs availability and accessibility in the community, respect for culture and tradition, problems regarding infrastructure, delay or unavailability of ambulance upon call, and some participants knowing only TBAs for birth assistance were reasons for preference of TBAs. Therefore, community-based interventions like health education on obstetric danger signs by giving more emphasis on the benefits of using ANC, strengthening access to transportation, health information communication through mass media will enhance the use of SBAs. Moreover, strengthening respectful maternity care services throughout the continuum of care could help to increase women's preference towards SBAs.

## Supporting information

**S1 File. English version of the questionnaire.**
(DOCX)

**S2 File. SPSS dataset.**
(SAV)

## Acknowledgments

We appreciate and wish to thank Debre Berhan University for providing ethical clearance to carry out this study. We also would like to acknowledge the Woreda administrators, data collectors, and study participants.

## Author Contributions

**Conceptualization:** Birhan Tsegaw Taye.

**Data curation:** Birhan Tsegaw Taye, Mulualem Silesh Zerihun, Tebabere Moltot Kitaw, Tesfanesh Lemma Demisse, Solomon Adanew Worku, Girma Wogie Fitie, Yeshinat Lakew Ambaw, Nakachew Sewnet Amare, Geremew Kindie Behulu, Addisu Andualem Ferede, Azmeraw Ambachew Kebede.

**Formal analysis:** Birhan Tsegaw Taye, Mulualem Silesh Zerihun, Tebabere Moltot Kitaw, Tesfanesh Lemma Demisse, Solomon Adanew Worku, Girma Wogie Fitie, Yeshinat Lakew Ambaw, Nakachew Sewnet Amare, Geremew Kindie Behulu, Addisu Andualem Ferede, Azmeraw Ambachew Kebede.

**Investigation:** Birhan Tsegaw Taye, Mulualem Silesh Zerihun, Tebabere Moltot Kitaw, Tesfanesh Lemma Demisse, Solomon Adanew Worku, Girma Wogie Fitie, Yeshinat Lakew Ambaw, Nakachew Sewnet Amare, Geremew Kindie Behulu, Addisu Andualem Ferede, Azmeraw Ambachew Kebede.

**Methodology:** Birhan Tsegaw Taye, Mulualem Silesh Zerihun, Tebabere Moltot Kitaw, Tesfanesh Lemma Demisse, Solomon Adanew Worku, Girma Wogie Fitie, Yeshinat Lakew Ambaw, Nakachew Sewnet Amare, Geremew Kindie Behulu, Addisu Andualem Ferede, Azmeraw Ambachew Kebede.

**Validation:** Birhan Tsegaw Taye, Mulualem Silesh Zerihun, Tebabere Moltot Kitaw, Tesfanesh Lemma Demisse, Solomon Adanew Worku, Girma Wogie Fitie, Yeshinat Lakew Ambaw, Nakachew Sewnet Amare, Geremew Kindie Behulu, Addisu Andualem Ferede, Azmeraw Ambachew Kebede.

**Visualization:** Birhan Tsegaw Taye, Mulualem Silesh Zerihun, Tebabere Moltot Kitaw, Tesfanesh Lemma Demisse, Solomon Adanew Worku, Girma Wogie Fitie, Yeshinat Lakew Ambaw, Nakachew Sewnet Amare, Geremew Kindie Behulu, Addisu Andualem Ferede, Azmeraw Ambachew Kebede.

**Writing – original draft:** Birhan Tsegaw Taye.

**Writing – review & editing:** Birhan Tsegaw Taye, Mulualem Silesh Zerihun, Tebabere Moltot Kitaw, Tesfanesh Lemma Demisse, Solomon Adanew Worku, Girma Wogie Fitie, Yeshinat Lakew Ambaw, Nakachew Sewnet Amare, Geremew Kindie Behulu, Addisu Andualem Ferede, Azmeraw Ambachew Kebede.

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
