## [Decision Letter · Decision Letter 0]

26 Aug 2022

PONE-D-22-02742Traditional birth attendant utilization at birth and associated factors among women who gave birth in the last two years preceding the survey in Angolella Tara District, EthiopiaPLOS ONE

Dear Dr. Taye,

Thank you for submitting your manuscript to PLOS ONE. After careful consideration, we feel that it has merit but does not fully meet PLOS ONE’s publication criteria as it currently stands. Therefore, we invite you to submit a revised version of the manuscript that addresses the points raised during the review process.

We look forward to receiving your revised manuscript.

Kind regards,

Vicente Sperb Antonello, MD, MSc, Phd

Academic Editor

PLOS ONE

Journal Requirements:

2. Please include a caption for figure 1.

5. Thank you for submitting the above manuscript to PLOS ONE. During our internal evaluation of the manuscript, we found significant text overlap between your submission and the following previously published works, some of which you are an author.

- https://pubmed.ncbi.nlm.nih.gov/20701762/

- https://journals.plos.org/plosone/article?id=10.1371%2Fjournal.pone.0246237

- https://www.ncbi.nlm.nih.gov/pmc/articlerender.fcgi?artid=3282603

-

The text that needs to be addressed involves the Abstract, Introduction, and Discussion.

Please revise the manuscript to rephrase the duplicated text, cite your sources, and provide details as to how the current manuscript advances on previous work. Please note that further consideration is dependent on the submission of a manuscript that addresses these concerns about the overlap in text with published work.

Additional Editor Comments (if provided):

After careful evaluation of the article and data from by reviewers, I understand that the article must pass through revision to be accepted into Plos One. All reviewers point out relevant issues such as methodological and grammatical writing information that should be adjusted.

Reviewers' comments:

Reviewer's Responses to Questions

**Comments to the Author**

1. Is the manuscript technically sound, and do the data support the conclusions?

Reviewer #1: Yes

Reviewer #2: Yes

Reviewer #3: Yes

Reviewer #4: Partly

Reviewer #5: Yes

2. Has the statistical analysis been performed appropriately and rigorously? 

Reviewer #1: Yes

Reviewer #2: Yes

Reviewer #3: Yes

Reviewer #4: No

Reviewer #5: Yes

3. Have the authors made all data underlying the findings in their manuscript fully available?

Reviewer #1: Yes

Reviewer #2: Yes

Reviewer #3: Yes

Reviewer #4: No

Reviewer #5: Yes

4. Is the manuscript presented in an intelligible fashion and written in standard English?

Reviewer #1: Yes

Reviewer #2: Yes

Reviewer #3: Yes

Reviewer #4: Yes

Reviewer #5: No

5. Review Comments to the Author

Reviewer #1: 1. The study presents the results of original research.

2. Results reported have not been published elsewhere.

3. Experiments, statistics, and other analyses are performed to a high technical standard and are described in sufficient detail.

4. Conclusions are presented in an appropriate fashion and are supported by the data.

5. The article is presented in an intelligible fashion and is written in standard English.

6. The research meets all applicable standards for the ethics of experimentation and research integrity.

7. The article adheres to appropriate reporting guidelines and community standards for data availability.

Reviewer #2: The topic is extremely relevant and is justified by the impact of childbirth care models on maternal mortality. Some questions and suggestions need to be made to further qualify the manuscript. First, about the third research question: "Why do women prefer TBAs"? In the results, about 31% of the sample used this modality. So, I was wondering why?In the introduction, TBAs are contextualized and presented as a prevalent modality of childbirth care, but why was there no preference for this model in your sample? So the question "Why do women prefer TBAs"? It didn't seem right to me. I suggest modifying or withdrawing this question, and also discussing the results in this perspective. n the results, 58.2% have a negative attitude towards TBAs. This percentage is in relation to the total sample or the 131 who used TBAs.

Reviewer #3: Childbirth occurred at home are dangerous or even life-threatening for the pregnant woman and newborns, especially those without the assistance of trained attendants. This manuscript is properly written and showed interesting findings. Sample size determination is reasonable. Also, the authors adequately listed and analyzed the limitation of their study. The authors provided valuable data and it is important to help governments, socialists and gynecologists in decreasing maternal mortality and fetal mortality and related diseases.

Reviewer #4: this article need some revision:

1. abstract is too long more than 200 words

2. utilization and prevalence are not suitable as keywords

3. Methode: there is no validity and reliability test for instrument???

4. ethical consideration: should write the letter number of ethical approval

5. result: table 3 is not complete , no p value in each varibles and as logistic regression analyzis it need to display the candidates that selected , then the multi variate analyzis of the selected candidates, end of the result should be the most infuenced factor toward the dependent variable.

6. discussion: 2nd sentence need reference.

7. discussion: page 9 , 1st paragraph : not clear statement ( Nigerian government or Ethiopia government because the result in this study is better than in Nigeria (31,5% versus 65%)

8. page 9: last sentence : this rationalization is not in accordance with the research results above.

9. page 10 : 2nd paragraph : The rationalization does not fit in the explanation here but is more appropriate in explaining the results that women who are pregnant for the first time use TBA services more.

10. Conclusion: Content is not enough conclusion , did not answer the objectives of the study , most of statement and paragraph is the conclusion are recommendation.

11. please revise table 3: no p value in each variables?? it need one column for p(significant value). Not clear multi variate analyzis result / logistic regression ( in the table should display the p value in each variable , follow by choosing the candidate for multy analyzis etc).

12. please check the references list some references have no publisher???URL?? when cited?

Reviewer #5: Traditional birth attendant utilization at birth and associated factors among women who

gave birth in the last two years preceding the survey in Angolella Tara District, Ethiopia

Please provide a grammatical native speaker review.

Introduction: Please provide a better writting to a better understanding; Ex.

"Although women believed that that complication might arise from TBA care (13), low maternal educational level, rural residence, poor family wealth index, unemployed status, having >5 living children, married, predominantly farmers, married to spouses who were farmers without formal education, user-friendly care from TBAs, easily

accessible, delivery environment familiar to the clients’ respect from TBAs for the religious beliefs of clients were significant factors for women to used TBAs." It is too long and of difficult comprehension.

Methods: Please provide inclusion and exclusion criteria more clearly.

Discussion is adequate, but I suggest to the authors to improve the strenghts information in this section (limitations of the study).

I miss information about improving educational information.

6. PLOS authors have the option to publish the peer review history of their article (what does this mean?). If published, this will include your full peer review and any attached files.

Reviewer #1: No

Reviewer #2: No

Reviewer #3: No

Reviewer #4: No

Reviewer #5: **Yes: **Vanice Ferrazza Zaltron

---

## [Author Response · Author response to Decision Letter 0]

10 Oct 2022

September 13, 2022

Manuscript PONE-D-22-02742

Response to Reviewers

First of all, we would like to acknowledge your constructive comments on and valuable improvements to our paper. We have incorporated the suggestions and comments in the revised manuscript and clearly show in tracked changes and a point-by-point response, please see below.

Response to Editor comments 

S.N Comments Responses

01 Please ensure that your manuscript meets PLOS ONE's style requirements, including those for file naming. The PLOS ONE style templates. Thank you very much for your genuine advice and we have revised the manuscript as per your kind recommendation; PLOS ONE style template.

02 Please include a caption for figure 1. Thank you, we have provided at the ed of the manuscript.

03 Please include captions for your Supporting Information files at the end of your manuscript, and update any in-text citations to match accordingly. Thank you, we have provided under ‘supporting information heading’.

04 Please review your reference list to ensure that it is complete and correct. If you have cited papers that have been retracted, please include the rationale for doing so in the manuscript text, or remove these references and replace them with relevant current references. Any changes to the reference list should be mentioned in the rebuttal letter that accompanies your revised manuscript. If you need to cite a retracted article, indicate the article’s retracted status in the References list and also include a citation and full reference for the retraction notice. Thank you for your kind recommendation. We have added four additional references (i.e. 10, 24, 25 and 27) in the revised version. These references are added to address the reviewers’ comments and during our essential update. 

05 Thank you for submitting the above manuscript to PLOS ONE. During our internal evaluation of the manuscript, we found significant text overlap between your submission and previously published works. Please revise the manuscript to rephrase the duplicated text, cite your sources. Thank you for pointing out this text overlap. We have revised the manuscript to rephrase the duplicated text, cite sources, and provide details on the advancement of knowledge by this research.

Response to Reviewer(s) comments 

Reviewer #2

S.N Comments Responses

01 The topic is extremely relevant and is justified by the impact of childbirth care models on maternal mortality. Some questions and suggestions need to be made to further qualify the manuscript. 

First, about the third research question: "Why do women prefer TBAs"? In the results, about 31% of the sample used this modality. So, I was wondering why? In the introduction, TBAs are contextualized and presented as a prevalent modality of childbirth care, but why was there no preference for this model in your sample? So the question "Why do women prefer TBAs"? It didn't seem right to me. I suggest modifying or withdrawing this question, and also discussing the results in this perspective. Thank you for your positive feedback and constrictive comments, we have accepted and modified in the revised manuscript.

02 n the results, 58.2% have a negative attitude towards TBAs. This percentage is in relation to the total sample or the 131 who used TBAs. Thank you, the figure (58.2%) is of all study participants. 

Reviewer #4

01 Abstract is too long more than 200 words Thank you for your comment. However, many journals suggest 350, 200 word counts for abstract, PLOS ONE submission guideline allow any length. There are no restrictions on word count, number of figures, or amount of supporting information. The word count for our abstract is 329.

02 utilization and prevalence are not suitable as keywords You have pointed out an important issue. We have corrected.

03 Methods: there is no validity and reliability test for instrument??? Thank you, we have incorporated in the revised manuscript.

04 Ethical consideration: should write the letter number of ethical approval Thank you very much, provided.

05 Result: table 3 is not complete, no p value in each varibles and as logistic regression analyzis it needs to display the candidates that selected, then the multi variate analyzis of the selected candidates, end of the result should be the most influenced factor toward the dependent variable. Thank you, dear reviewer, we have provided it. However, p-values simply provide a cut-off beyond which we assert that the findings are 'statistically significant' (by convention, this is p<0.05). Confidence intervals provide information about statistical significance, as well as the direction and strength of the effect.

06 Discussion: 2nd sentence need reference. Comment accepted and corrected.

07 Discussion: page 9, 1st paragraph: not clear statement (Nigerian government or Ethiopia government because the result in this study is better than in Nigeria (31,5% versus 65%) Thank you, the Nigerian government encourage TBAs service integration with skilled birth attendants for childbirth. As a result, 65% TBA utilization at birth. Ethiopian gov’t is better (31.5% TBA utilization).

08 page 9: last sentence: this rationalization is not in accordance with the research results above. Really thank you for your effort. Corrected. Please kindly see the tracked changes.

09 page 10: 2nd paragraph: The rationalization does not fit in the explanation here but is more appropriate in explaining the results that women who are pregnant for the first-time use TBA services more. Your kind suggestion is accepted.

10 Conclusion: Content is not enough conclusion, did not answer the objectives of the study, most of statement and paragraph is the conclusion are recommendation. Thank you, all relevant findings incorporated.

11 please revise table 3: no p value in each variable?? it needs one column for p (significant value). Not clear multi variate analyzis result / logistic regression (in the table should display the p value in each variable, follow by choosing the candidate for multy analyzis etc). Comment accepted and corrected. 

12 please check the references list some references have no publisher???URL?? when cited? Thank you, some references amend accordingly. 

Reviewer #5

(Dr. Vanice Ferrazza Zaltron)

01 Please provide a grammatical native speaker review. Thank you for the comment, the language was reviewed exhaustively. It is also clearly visualized via track changes in the revised manuscript.

02 Introduction: Please provide a better writting to a better understanding; Ex.

"Although women believed that that complication might arise from TBA care (13), low maternal educational level, rural residence, poor family wealth index, unemployed status, having >5 living children, married, predominantly farmers, married to spouses who were farmers without formal education, user-friendly care from TBAs, easily

accessible, delivery environment familiar to the clients’ respect from TBAs for the religious beliefs of clients were significant factors for women to used TBAs." It is too long and of difficult comprehension. Thank you for your deep review, paraphrased.

03 Methods: Please provide inclusion and exclusion criteria more clearly. Thank you, provided.

04 Discussion is adequate, but I suggest to the authors to improve the strenghts information in this section (limitations of the study). I miss information about improving educational information. Thank you, provided.

---

## [Editor Report · Decision Letter 1]

31 Oct 2022

Women’s traditional birth attendant utilization at birth and its associated factors in Angolella Tara, Ethiopia

PONE-D-22-02742R1

Dear Dr. Taye,

We’re pleased to inform you that your manuscript has been judged scientifically suitable for publication and will be formally accepted for publication once it meets all outstanding technical requirements.

Kind regards,

Vicente Sperb Antonello, MD, MSc, Phd

Academic Editor

PLOS ONE

Additional Editor Comments (optional): I just suggest changing the section title "limitations of the study" to "Strengths and limitations of the study"

---

## [Editor Report · Acceptance letter]

3 Nov 2022

PONE-D-22-02742R1 

Women’s traditional birth attendant utilization at birth and its associated factors in Angolella Tara, Ethiopia 

Dear Dr. Taye:

I'm pleased to inform you that your manuscript has been deemed suitable for publication in PLOS ONE. Congratulations! Your manuscript is now with our production department. 

Kind regards, 

on behalf of

Dr. Vicente Sperb Antonello 

Academic Editor

PLOS ONE